# Cleaner wrasse *Labroides dimidiatus* perform above chance in a "matching-to-sample" experiment

**Mélisande Aellen** [1]*, **Ulrike E. Siebeck**[2], **Redouan Bshary**[1]

**1** Department of Behavioural Ecology, University of Neuchâtel, Neuchâtel, Switzerland, **2** School of Biomedical Sciences, University of Queensland, Brisbane St Lucia, QLD, Australia

* melisande.aellen@outlook.com

**Data Availability Statement:** All data and the R-scripts to reproduce the analyses are available at https://figshare.com/s/4ed4d7c5bcc750500de9.

**Funding:** The study was supported by the Swiss National Science Foundation (grant numbers

## Abstract

Concept learning have been studied widely in non-human animal species within or not an ecological context. Here we tested whether cleaner fish *Labroides dimidiatus*, which show generalised rule learning in an ecologically relevant context; they generalise that any predator may provide protection from being chased by other fish; can also learn a general concept when presented with abstract cues. We tested for this ability in the matching-to-sample task. In this task, a sample is shown first, and then the subject needs to choose the matching sample over a simultaneously presented different one in order to obtain a food reward. We used the most general form of the task, using each stimulus only once in a total of 200 trials. As a group, the six subjects performed above chance, and four individuals eventually reached learning criteria. However, individual performance was rather unstable, yielding overall only 57% correct choices. These results add to the growing literature that ectotherms show the ability of abstract concept learning, though the lack of stable high performance may indicate quantitative performance differences to endotherms.

## Introduction

Abstract concept learning show the ability to categorize/classify objects based on similar shape, association or relation equities and then being able to transfer this knowledge to new conditions/situations [1]. Four main different types of abstract concept learning are described, differing in the way individuals have to discriminate between stimuli (reviewed in [2]). Discrimination can be based on forming classes, such as chair or flower, known as perceptual concept learning [3]; discrimination to form categories by associating stimuli to another, such as an object with the word for that object, known as associative concept learning (reviewed in [4]); discrimination via the relationship between or among stimuli, such as same/different, known as the relational concept learning [5]; and finally the discrimination via analogy, i.e. develop the relations between relations, such as a set of five same icons is similar to a set of five same but different icons and dissimilar to a set of five different icons, known as the analogical reasoning [6–8]. Overall, abstract concepts learning are represented by the relation of taught

31003A_153067 and 310030B_173334/1) to R.B.
The funders had no role in study design, data
collection and analysis, decision to publish, or
preparation of the manuscript.

rules with training stimuli to novel stimuli. Because such cognitive flexibility is supposedly more complex than standard reinforcement learning [9–12], it has attracted considerable attention in animal cognition research [13].

As the ability to form abstract concepts is considered to be cognitively demanding, only large-brained animal species, like primates, were initially taken into consideration [10, 14–17]. However, the ability to learn abstract concepts eventually turned out to be widespread in the animal kingdom, like in primates, dolphins, seals, birds, rodents, octopus, mollusks, insects and fish [2, 18–20]. Experiments on abstract concept learning test for the animals' ability to generalise. Generalisation requires the ability to conceptualise a new environmental input based on available knowledge, i.e. relating a new situation with information acquired from previous experience [21]. A standard approach is to offer subjects in succession diverse pairs of stimuli, where only one is rewarding. Each single combination can be learned by operant conditioning, i.e. the subject associating its choice of stimulus with either a positive or a negative reinforcement [22, 23]. However, if subjects are able to extract the general rule that "in any pair of stimuli, one is rewarding while the other one is not", they can solve any new stimulus combination from the second trial onwards. This is because while the choice during the first trial is necessarily random, a reward shows that the chosen stimulus is the correct one, while no reward shows that the non-chosen stimulus must be the correct one [24]. Another commonly used task is the same/different discrimination where individuals have to discriminate if two stimuli are the same or if they are different based on their physical pattern [25].

Yet another paradigm employed to determine the relational, i.e. same/different, concept competence in animals is the matching-to-sample task (hereafter MTS) [26–30]. Subjects are shown a stimulus (the sample), and then they can choose between simultaneously presented matching and differing stimuli. Only the choice of the matching stimulus is rewarded. In the simultaneous MTS, the sample stimulus is presented and then both, the matched and the non-matched stimuli are presented beside it [31–36]. In the delayed MTS, the sample is removed before the two choice stimuli are presented [36]. Crucially, the sample varies from trial to trial, forcing subjects to comprehend the concept of 'same' and 'different' by relating the choice stimuli to the sample in order to succeed [1, 2]. Hitherto, two main experimental paradigms have been used to test animals. In the most commonly used paradigm, subjects are confronted during an initial learning phase repeatedly with the same two choice items but each is presented as sample stimulus in a counterbalanced way [1, 15, 32, 36, 37]. Once subjects reach learning criterion (often only after running hundreds of trials), a transfer test is conducted: two new stimuli are introduced and it is tested whether subjects reach learning criterion faster with the new combination (evidence for transfer of acquired knowledge). In the other paradigm, experimenters operate from the beginning with a larger number of samples and corresponding pairs of stimuli [37].

Studies using the transfer task paradigm showed variation in the performance of the different taxa tested, with several studies yielding positive results (bottlenose dolphins [38, 39], California sea lions [40], capuchin monkeys [15, 33], chimpanzees ([36, 37]; cited by [41]), hens [42], honeybees [43], pigeons [29, 32, 37, 44], and rats [31]; summarized in S1 Table). Different senses were used in those diverse studies, such as olfaction, audition, and vision. Overall, only a few studies have tested fish so far: goldfish [45, 46], Malawi cichlids [47], archerfish [35], and zebrafish [36]. In those studies, all fish were tested with visual stimuli using either coloured light or forms and shapes. Evidence for simultaneous MTS was found in goldfish [46] and zebrafish [36], but neither in cichlids [47] nor in archerfish [35]. No fish species has yet been tested in the second paradigm which uses a large number of different samples over successive trials. This is unfortunate as the second paradigm is more indicative of generalised rule

learning, as the transfer task paradigm may yield high performance with subjects being able to 'learn-how-to-learn' [44, 48] rather than using a general rule.

Given the mixed evidence for fishes regarding their capacity for conceptual learning in the simultaneous MTS task, we tested cleaner fish *Labroides dimidiatus*, using different shapes and colour 2D images as stimuli, where any stimulus was used only in one trial. Cleaner fish are a suitable study species for an MTS task using abstract stimuli as there is evidence that they use generalised rule learning in an ecologically relevant context [49]. The ecologically relevant context is linked to conflicts between cleaners and 'client' reef fish that arise when cleaner feed on client mucus rather than on client ectoparasites [50]. Clients sometimes respond to cleaners eating mucus with aggressive chasing [51]. In such situations, cleaners may seek a passing-by predatory client, which functions as a social tool as its mere proximity makes the punishing individual stop its pursuit [52]. Within this context, Wismer et al. (2016) showed evidence for generalised rule learning in controlled laboratory experiments. In these experiments, cleaners were offered preferred and non-preferred food items on a Plexiglas plate. In response to cleaners eating a preferred item, the experimenter would start chasing the cleaner with the plate (manoeuvring a lever attached to the plate). The chasing would push the cleaners in the direction of two laminated pictures at the end of the aquarium, one showing a predatory client and one showing a harmless client. If the cleaner approached the picture of the predator, the pursuit with the plate stopped, while the pursuit continued if the cleaner moved towards the image of the harmless client. Once individual cleaners showed a significant preference for the predator picture, the two pictures were exchanged. In these follow-up tasks, cleaners reached learning criteria significantly faster, which can interpreted as evidence for generalised rule learning [49], albeit with the caveat that 'learning to learn' offers an alternative explanation.

The results fit the prediction that animals typically excel in tasks that are presented in an ecologically relevant context [53, 54]. Solving ecologically relevant tasks may be achieved by any animal species, while more abstract presentations supposedly warrant relatively large brains that allow flexibility beyond concise situations [55–58]. We, therefore, wanted to test if cleaner fish are capable of demonstrating the ability of concept learning even when the task has no ecological relevance, i.e. in an abstract MTS paradigm. In order to avoid any 'learn-how-to-learn' [44, 48] explanations as in the Wismer et al. (2016) study, and indeed all previous studies providing positive results on the MTS task, we opted against any repeated use of stimuli. Studies using multiple sample stimuli suggest that if anything the use of more stimuli (up to 76 in pigeons) may enhance generalised rule learning [32, 33, 59, 60]. Thus, in contrast to previous studies on MTS, we ran no training trials prior to testing. Instead, each symbol was used only once per fish. Under these circumstances, any performance above chance must be based on the formation of a general concept. The experiment should answer the question of whether cleaner fish show evidence for generalised rule learning only within a narrow ecologically relevant context or whether they can form concepts also in more abstract circumstances. In the latter case, we would conclude that concept learning is a domain-general cognitive capacity of cleaner fish.

## Materials and methods

This study was conducted from April until May 2017 at the Gump research station on Moorea (French Polynesia). Six cleaner fish (*Labroides dimidiatus*) between 7.5 and 8.6 cm were caught with a barrier net (2m long, 1.5m high, mesh size 0.5cm) and using hand-nets in lagoons around Moorea. Cleaners were accustomed to plastic aquaria of a size of 69 x 50 x 43 cm (S1 Fig), the water level at 34 cm, with a small PVC pipe as a shelter for at least 19 days before experiments started. A pipe from the lagoon provided a continuous seawater flow.

### Acclimation and pre-training

After seven days in captivity, cleaners were first trained to feed small items of mashed prawn off small dots painted on small Plexiglas plates (10 x 7 cm). For the experiment, they were habituated to approach the experimental plate (20 x 20 cm) with a random training symbol in the middle (symbol drawn on a white square piece of paper of dimension 3 x 3 cm), and when they would touch or bite the training symbol they would receive a food reward, i.e. a piece of mashed prawn, by inserting the reward plate (3 cm width). As a first training step, a food item was placed directly on the training symbol itself. Once fish were comfortable to approach the training symbol, the reward plate was used. Furthermore, subjects were familiarized with a dashed see-through barrier placed 10 centimetres from the aquarium wall with a sliding door inserted in the middle. The barrier created an "experimental" compartment into which the plate was introduced and a "resting" compartment to which the cleaner was initially confined. The door was opened once the experimental plate had been put into place (Fig 1). All cleaners readily approached the experimental plate before the trials began. The time it took each individual to be accustomed to the experimental setup and hence to start the experiment varied between 11 and 17 days. Consequently, not all individuals had the same amount of training days, and thus they did not start the experiment on the same day.

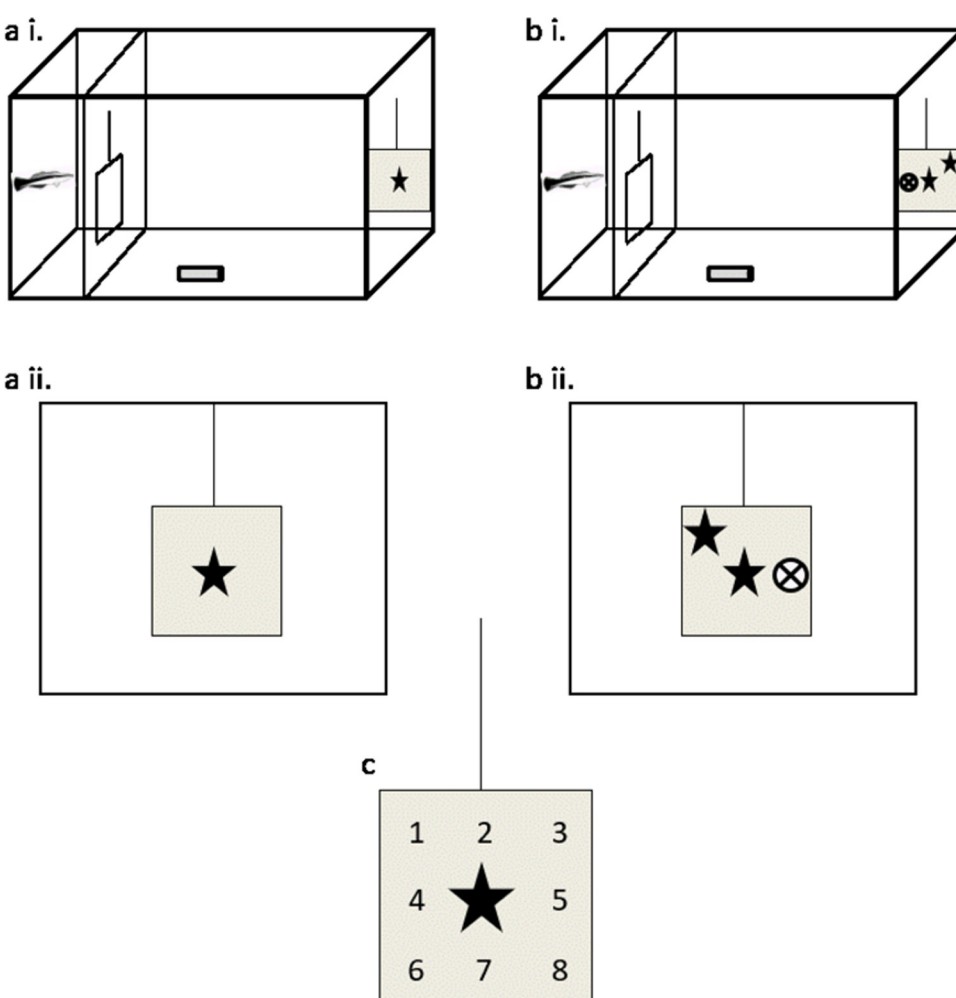

**Fig 1.** Exposition 1 (a) and 2 (b) of the experimental plate. i. Set up, ii. Fish view. We used eight different positions around the central sample symbol as emplacements for the matched and the non-matched symbols (c).

## Experimental procedure

Cleaners were tested in their home aquarium. Once the fish was in the "resting" area, we fixed a GoPro camera using a clamp to clip it at the edge of the aquarium wall such that it filmed the trial. After 60 seconds, we introduced the experimental plate with the "sample" symbol placed in the middle of a 3x3 square matrix with 5 cm between symbols emplacement, for about 10 seconds (giving the fish time to look at the "sample" symbol; Fig 1A). The symbols varied in shape, colour, and were of a dimension of 3x3 cm and were created for the purpose of this study. The experimental plate, with hooks glued on the back, was hanged on the aquarium wall avoiding hand movement that might interfere with the fish's choice. The plate was then removed to add the "match" and "non-match" symbols. To avoid the development of side bias, these two symbols could be placed on any of the eight remaining squares (Fig 1C), with the exact locations pre-determined to balance positions across trials. After 25–30 seconds, the experimental plate was re-inserted with the three symbols (Fig 1B). The door was opened 3–5 seconds later so that the cleaner could make its choice. The fish had to be in physical contact with a symbol (either by giving tactile stimulation with its pelvic fins or by touching it with the mouth) so that we considered that a choice had been made. Only if the choice was the matching symbol, the cleaner received a small piece of mashed prawn as a food reward on the introduced reward plate. When it had finished eating, we gently removed both the rewarding and the experimental plate. If the "non-match" symbol was chosen, no reward plate was introduced but instead, the experimental plate was quickly removed. We completed one trial with each fish before we conducted the next round of trials. As a result, the intertrial interval (ITI) was about twenty minutes for each fish. Each individual fish was exposed to a maximum of 200 trials, each one presenting two new symbols, so that the fish was confronted with a maximum of 400 different symbols. The same set of 400 symbols was used for all fish but combinations, locations, and function ("match" and "not-match" symbols) were randomised (see S2 Fig for the first 20 symbols used, the rest of the symbols utilized can be found on 10.6084/m9. figshare.13522100).

Ten trials were conducted per individual per day. As long as cleaners did not reach the confirmation criteria (explained in the next paragraph), they were continuing in the experiment until the completion of 200 trials, i.e. 20 experimental days. On a few occasions, fish did not make a proper choice, i.e. they randomly touched both symbols with their body by swimming over them without stopping, and swimming back and forth in the aquarium until the 60s trial was timed out. These trials were discarded but not replaced. As a consequence, few individual daily data sets contained only nine data points. We directly recorded the individuals' choices (correct/incorrect) but all trials were also video recorded and can be requested. Fish were fed for five minutes 20 minutes prior to the start of the experiment. As it stands cleaner fish eat regularly over eleven hours per day, and hence are likely hungry in the morning after 13 hours of fasting. Feeding them prior to the experiment aimed at reducing the probability that they make random choices because of high hunger levels. During the trials, cleaners did not obtain much food (ten small prawn items max if all choices had been correct). Therefore, when the experiment was finished for the day, the food plate was introduced for about two hours so that subjects could obtain the caloric needs for the day.

## Learning and confirmation criterion

An individual was considered to have learned the task if it performed significantly above chance levels (based on a sign test table/ binomial tests) in either one session (9 or 10 out of 10 trials correct), two successive sessions (twice at least 8 out of 10 trials correct) or three consecutive sessions (three times at least 7 out of 10 trials correct). All these learning criteria yield a

p < 0.05 that performance was due to chance. In order to be conservative, we exposed any individual that reached a learning criterion to another 10 trials and only considered the individual to have succeeded if it chose correctly at least 7 out of 10 trials. Failure to achieve at least 7 out of 10 correct choices meant that the individual continued the simultaneous MTS experiment.

## Statistical analyses

Apart from the individual learning criteria, we also asked whether our six cleaners as a group performed above chance levels, i.e. > 50%. We ran a fit Bayesian Generalized Linear Mixed-Effects Models with the statistical program Rstudio © (R Version 1.3.1093, © 2009–2019 RStudio, PBC) [61] using the package 'lme4' [62] and 'blme' [63]. The individuals' choice was set as a binary response variable. In order to verify whether the session had an effect in our model, we set it as a fixed-effect variable. The individual was set as the random-effect variable (both at intercept and at slope level with respect to trial) of the 'bglmer' model. All data and the R-scripts to reproduce the analyses are available on 10.6084/m9.figshare.13522100.

## Ethics approval

Our research study adheres to the ASAB/ABS Guidelines for the Use of Animals in Research. The study was approved by the French Polynesian authorities responsible of the program 'Délégation à la Recherche' and by the Gump Research Station where the study was conducted in accordance with their rules and regulations for animal research. We acknowledge that catching is a stressful event for the fish. Afterwards, the fishes adapt well to captive conditions and lose their shyness towards human experimenters. The cognitive experiments can be seen as behavioural enrichment. All fish were released at their site of capture.

## Results

Four out of the six cleaners eventually reached the learning criterion. Two of these four cleaners also reached the confirmation criterion in the next session, while the other two failed and never reached the learning criterion again until the completion of the 200 trials (Fig 2). Of the

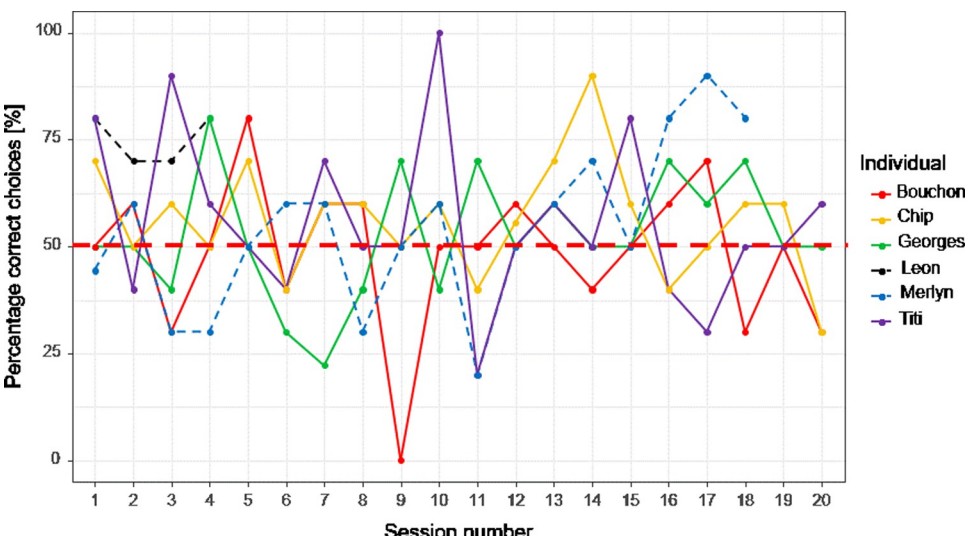

**Fig 2. Simultaneous matching-to-sample experiment.** The percentage when the matched symbol was chosen is represented as function of the session number (1 session = 10 trials; n = 6). The red dashed horizontal line indicates chance expectation (50% correct choices).

**Table 1. Mean percentage of correct choice (= success) and standard deviation of each individual throughout all sessions tested.**

|  | Bouchon | Chip | Georges | Leon | Merlyn | Titi |
|---|---|---|---|---|---|---|
| **Mean % success** | 48.5% | 56.3% | 52.61% | 75% | 54.08% | 56% |
| **Standard deviation** | 17.3 | 13.5 | 14.6 | 5.8 | 18.9 | 19.8 |

two successful cleaners, one completed the task in 40 trials (including the confirmation session), while the other one needed in total 180 trials.

Combining individual performances, cleaners succeeded significantly above chance as a group (estimate = 0.18, std. error = 0.07, z-value = 2.51, p = 0.01; Fig 2). The effect size is small (n = 6, mean = 57.08%). Individual mean performance varied between 48.5% and 75% (Table 1). This latter high value was due to the individual reaching criterion within only 40 trials. Even without that individual, cleaners still performed above chance as a group (estimate = 0.15, std. error = 0.07, z-value = 2, p = 0.046). Nevertheless, we note that overall mean performance (57%) was only slightly above chance level, and there was no significant improvement with experience as the factor 'session' was not significant (estimate = - 0.0002, std. error = 0.01, z-value = - 0.01, p = 0.99).

## Discussion

We had asked to what extent cleaner wrasses, *Labroides dimidiatus*, were able to learn an abstract general rule without experiments that allow the acquisition of intermediate steps, using the MTS task. According to our results, cleaners have the potential to learn an abstract rule as they performed above chance as a group. Four out of six cleaners eventually reached learning criteria that were set such that individuals performed better than expected by chance ([64]; described in [65, 66]), and two of these cleaners even passed a more conservative criterion. However, the cleaners' performance was also not consistent, with two individuals returning to chance levels after reaching the initial criterion. This contrasts with other experiments in which cleaners generally continued to perform well in sessions after reaching learning criteria ([67]; Salwiczek et al., unpublished data). It also contrasts with the cleaners' consistent ability to use generalised rule learning in an ecologically relevant task [49]. Taken together, as a species, cleaners expressed at best some limited understanding, with only a minority performing above chance. Single individuals excelling at specific tasks is a widespread phenomenon in cognitive studies [68–71], highlighting the importance to understand intraspecific variation in cognitive performance [72].

To the best of our knowledge, the procedure that we used, i.e. the simultaneous MTS task in its most general form, was never tested before in non-human animals. Most previous studies started with a training phase that consisted of using two stimuli that alternated as matched and non-matched symbols, referred to as the 'standard MTS paradigm'. When an individual reached learning criterion, another set of stimuli known as the transfer stimuli was presented. This transfer task was introduced to rule out the possibility that subjects had learned two configurations rather than the rule 'match to sample'. If individuals had learned the rule, the number of trials needed to reach criterion with those second transfer stimuli should decrease compared to the training stimuli. While this approach yielded many positive results (S1 Table), some authors proposed the alternative explanation that subjects potentially merely learned how to learn rather than understand the concept [2, 18, 20, 44, 48]. There is some evidence that the general version may be more challenging than the standard version of the MTS task. Training pigeons with 76 trials per day, Wright et al. (1988) exposed two subjects to the standard version (2 stimuli alternating roles) and 2 subjects to 76 fixed pairs of stimuli (that

would be used again over consecutive days, alternating roles as well). The two pigeons exposed to the standard version reached criteria after > 1000 trials, while it took the other two pigeons > 27000 trials. On the other hand, the latter performed very well in transfer trials while the former did not. Given that small differences in experimental design may cause major differences in performance, we consider it important that more studies will use our design on a variety of species belonging to different major vertebrate clades. In return, future research on cleaner fish should use the transfer task paradigm for the sake of obtaining data that are comparable to existing studies on MTS.

The fact that we obtained overall performance slightly above chance levels despite using new stimuli at every trial, as well as the significant performance of two individuals, favours the interpretation that cleaner wrasse are indeed capable of abstract concept learning [73, 74]. However, we note that the fish did not improve as a function of session their performance from the first 100 trials to the second 100 trials, which would be expected if they learned the task. One possibility is that the increment of the number of stimuli used during the experiment increased the information load that the fish were exposed to [73]. The constant use of new stimuli may have caused an overload of information, which increased the cognitive difficulty of the task and prevented a continued improvement of performance over the trials. It has been documented before that more information does not necessarily lead to better performance [74, 75]. Alternatively, cleaners might have a small perception bias that causes them to slightly favour the symbol that they see twice. To rule out that potential explanation, future experiments should reward the symbol that is the 'Oddity from sample' [76]. If cleaners perform above chance level in this opposite reward scheme, perception biases could be ruled out as an explanation for cleaner performance. The available literature suggests that choosing the odd option is simpler than choosing the matching option [76, 77], even more in the delayed version of the task [78–80].

Until now, positive evidence for abstract rule learning by fishes in the MTS task was restricted to goldfish [45, 46] and zebrafish [36], the proper controls conducted with a negative reinforcer (electric shock) [46]. In contrast, archerfish [35] and a cichlid (*Pseudotropheus* sp.) failed at MTS tasks [47]. The current study with cleaner fish thus provides some additional evidence that fish (at least some individuals) can learn an abstract concept. We cannot propose an explanation for the positive results for goldfish and cleaners and the negative results for the other two species. The key idea of the MTS task is that it tests for abstract concept learning in a context that lacks ecological relevance. Neither goldfish nor cleaners appear to have particularly large brains compared to other fish species (Allen et Kuiter, 1999; Cuvier et Valenciennes, 1839; Fowler et Bean, 1928; Lacépède, 1801; Randall, 1981; Bleeker, 1855; all cited in [81, 82]). At least for cleaner fish there is evidence that they are able to use generalized rule learning in an ecologically relevant context [49, 52].

## Conclusions

Cleaner wrasses, *L. dimidiatus*, can solve a MTS task but individual performance is not consistent, at least in the general version we used. We cannot compare their performance to other species because of the new design. In future studies, it would be interesting to compare the standard design (training with limited set of stimuli prior to continuously novel stimuli) with our design (continuously novel stimuli only) in a diversity of species, including cleaners. Compared to the data showing that pigeons require many thousands of trials with novel stimuli until they learned the task, the performance of cleaners within the 200 trials presented here was rather good. Thus, generalised rule learning in cleaners does not seem to be restricted to ecologically relevant contexts, indicating that abstract concept learning is a general cognitive tool present in this species.

## Supporting information

**S1 Fig. Experimental aquarium design.** The "resting" area is on the left where the fish is waiting behind a see-through grid barrier with a door in the middle. On the right is the "experimental" compartment where the experimental plate is displayed during the trial.
(TIF)

**S2 Fig. Representation of the twenty first symbols used in the matching-to-sample task.** The symbols were drawn on a 3 x 3cm white paper sheet.
(TIF)

**S1 Table. Main results of MTS studies.**
(PDF)

## Acknowledgments

We are grateful to the Gump research station in Moorea (French Polynesia) for letting us work in their station. We are particularly thankful to the directors and the staff for their unconditional help and great friendship. We thank Léonore Bonin for fish capturing. We acknowledge Dr. Christèle Borgeaud for English corrections and valuable comments. We thank Radu Alexandru Slobodeanu for statistical help and support.

## Author Contributions

**Conceptualization:** Mélisande Aellen, Redouan Bshary.

**Data curation:** Mélisande Aellen.

**Formal analysis:** Mélisande Aellen.

**Investigation:** Mélisande Aellen.

**Methodology:** Mélisande Aellen, Ulrike E. Siebeck, Redouan Bshary.

**Project administration:** Mélisande Aellen.

**Resources:** Redouan Bshary.

**Supervision:** Redouan Bshary.

**Validation:** Mélisande Aellen.

**Visualization:** Mélisande Aellen.

**Writing – original draft:** Mélisande Aellen.

**Writing – review & editing:** Mélisande Aellen, Ulrike E. Siebeck, Redouan Bshary.

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
