## [Decision Letter · Decision Letter 0]

14 Jul 2021

PONE-D-21-17431

Cleaner wrasse Labroides dimidiatus perform above chance in a "Matching-to-sample" experiment

PLOS ONE

Dear Dr. Aellen,

Thank you for submitting your manuscript to PLOS ONE. After careful consideration, we feel that it has merit but does not fully meet PLOS ONE’s publication criteria as it currently stands. Therefore, we invite you to submit a revised version of the manuscript that addresses the points raised during the review process.

So far I have had one independent expert on animal behaviour review the manuscript, and I have also carefully read the work. I agree with R1 that the work is potentially of interest to publish, but clarity of explanations and writing can be improved a lot (see reveiwer report). In particular, I agree with R1 that readers need a more complete explanation of MTS (matching to sample)for which you use simultaneous presentation, and how delayed MTS (ie DMTS) experiments has been more frequently used in other animal species, at least in recent times. DMTS requires buth longer term memory formation of rule, and working memory (WM) of samples in a particular instance. The simultaneous MTS thus is not directlly comparable to DMTS; at least you need to clearly explain differences for readers and how such differences influence your experimental interpretation(s).

I also found the line "Fish were fed ad libitum for five minutes 20 minutes prior to the start of the experiment." confusing as to what this meant.........please use a lot more care so a general readership can fully understand your paper. I also was confused by s3 Figure which I cant see mentioned in main manuscript. Can you please take much more care in paper presentation and message? I have allowed time for a major revision to enable this. If you can do a good job I will then ask R1 and another expert to review the revised paper.

We look forward to receiving your revised manuscript.

Kind regards,

Adrian G Dyer, Ph.D.

Academic Editor

PLOS ONE

Journal Requirements:

 [The study was supported by the Swiss National Science Foundation (grant numbers 31003A_153067 and 310030B_173334/1) to R.B. 

NO]. 

a) Please provide an amended Funding Statement that declares *all* the funding or sources of support received during this specific study (whether external or internal to your organization) as detailed online in our guide for authors at http://journals.plos.org/plosone/s/submit-now.  

b) Please state what role the funders took in the study.  If any authors received a salary from any of your funders, please state which authors and which funder. If the funders had no role, please state: "The funders had no role in study design, data collection and analysis, decision to publish, or preparation of the manuscript." 

Please send your amended statements by return email; we will change the online submission form on your behalf. 

[We are grateful to the Gump research station in Moorea (French Polynesia) for letting us work in their station. We are particularly thankful to the directors and the staff for their unconditional help and great friendship. We thank Léonore Bonin for fish capturing. We acknowledge Dr. Christèle Borgeaud for English corrections and valuable comments. We thank Radu Alexandru Slobodeanu for statistical help and support. The study was supported by the Swiss National Science Foundation (grant numbers 31003A_153067 and 310030B_173334/1) to R.B.]

Please remove any funding-related text from the manuscript and let us know how you would like to update your Funding Statement. 

Reviewers' comments:

Reviewer's Responses to Questions

**Comments to the Author**

1. Is the manuscript technically sound, and do the data support the conclusions?

Reviewer #1: Yes

2. Has the statistical analysis been performed appropriately and rigorously? 

Reviewer #1: I Don't Know

3. Have the authors made all data underlying the findings in their manuscript fully available?

Reviewer #1: No

4. Is the manuscript presented in an intelligible fashion and written in standard English?

Reviewer #1: No

5. Review Comments to the Author

Reviewer #1: In this study, the authors tested the ability of cleaner fish to succeed in a simultaneous matching-to-sample task. Such task is classically used to test for rule learning based on a ‘same’ abstract relational concept. This cognitive ability has been established in many animal species from mammals to insects. However, while some fish species appeared able to master this task (zebrafish and goldfish), others were not (e.g. archerfish). The current study thus proposes to further explore fish concept learning capacities by testing an additional species. This question is scientifically valid and of interest as I consider that the comparative cognition field certainly gains from more model species to be tested for key cognitive tasks. The results demonstrate great individual variability but with success of a few individuals. These results support the existence of an ability for solving matching-to-sample task in this species in particular as the procedure chosen is not classic and may have had detrimental effect on performance. Indeed, usually, the sample is not longer present when the subject has to make a choice between a stimulus identical to the sample and an alternative stimulus. In addition, only a limited number of samples are used in standard procedures at least during the acquisition phase while in the current study, each trial presented a novel stimulus. These key differences may have induced attentional bias and noise which prevented most subjects to extract the common rule between trials. Proposing alternative procedures is of high interest to better understand the underlying processes but I regret that the authors did not test in parallel the classical procedure for comparison. With the current data, it is indeed impossible to compare the cleaner fish performances with those of other species.

Other comments:

Introduction: The introduction needs quite a substantial editing. Some sentences are difficult to read, some others are incorrect or imprecise. I recommend the authors to invest more effort in drafting the introduction in a novel version of the manuscript.

A few examples:

l. 35 : do you mean a shaping procedure ? It is not clear what you mean by operant conditioning. Your description does not allow to understand what is the procedure to study the ability of animals to learn abstract concepts. Please revise the lines 34-37

l. 46 : I do not understand this sentence. ‘concepts are represented by the relation of abstract rules to novel stimuli’ ??

l. 49-53 : Difficult to follow, please revise for improved clarity

l.57 : The reference 27 seems fully out of context here. Could you please provide references of studies using simultaneous MTS to test for same/difference concept learning ?

l. 58: Incorrect references as they used a DMTS procedure and not a simultaneous MTS

…

Methods

l. 164 : Why did you feed the fish ab libitum before the experiment. This should induce a lack of motivation for the food reward.

l. 165 : Why did you let the food plate in place for 2 hours ?

I suggest to provide a video as supporting information to help the reader to better picture the whole procedure.

Discussion

l.228 : There is at least one study, the one in pigeons which is described a few sentences later (ref 12)

Bibliography : please check the references formatting as not consistent between references and does not follow the journal guidelines.

6. PLOS authors have the option to publish the peer review history of their article (what does this mean?). If published, this will include your full peer review and any attached files.

Reviewer #1: No

---

## [Author Response · Author response to Decision Letter 0]

28 Sep 2021

CC: adrian.dyer@rmit.edu.au

PONE-D-21-17431

Cleaner wrasse Labroides dimidiatus perform above chance in a "Matching-to-sample" experiment

PLOS ONE

Dear Dr. Aellen,

Thank you for submitting your manuscript to PLOS ONE. After careful consideration, we feel that it has merit but does not fully meet PLOS ONE’s publication criteria as it currently stands. Therefore, we invite you to submit a revised version of the manuscript that addresses the points raised during the review process.

So far I have had one independent expert on animal behaviour review the manuscript, and I have also carefully read the work. I agree with R1 that the work is potentially of interest to publish, but clarity of explanations and writing can be improved a lot (see reveiwer report). In particular, I agree with R1 that readers need a more complete explanation of MTS (matching to sample)for which you use simultaneous presentation, and how delayed MTS (ie DMTS) experiments has been more frequently used in other animal species, at least in recent times. DMTS requires buth longer term memory formation of rule, and working memory (WM) of samples in a particular instance. The simultaneous MTS thus is not directlly comparable to DMTS; at least you need to clearly explain differences for readers and how such differences influence your experimental interpretation(s).

I also found the line "Fish were fed ad libitum for five minutes 20 minutes prior to the start of the experiment." confusing as to what this meant.........please use a lot more care so a general readership can fully understand your paper. I also was confused by s3 Figure which I cant see mentioned in main manuscript. Can you please take much more care in paper presentation and message? I have allowed time for a major revision to enable this. If you can do a good job I will then ask R1 and another expert to review the revised paper.

Dear Dr Dyer, we would like to thank you for your comments. We considered each input and hope that the new version will be more suitable and reach your journal standards in PLOS ONE.

We look forward to receiving your revised manuscript.

Kind regards,

Adrian G Dyer, Ph.D.

Academic Editor

PLOS ONE

Journal Requirements:

Thank you, we will carefully follow the journal guidelines.

 [The study was supported by the Swiss National Science Foundation (grant numbers 31003A_153067 and 310030B_173334/1) to R.B. 

NO]. 

a) Please provide an amended Funding Statement that declares *all* the funding or sources of support received during this specific study (whether external or internal to your organization) as detailed online in our guide for authors at http://journals.plos.org/plosone/s/submit-now.

We took good notes of it and will do the necessary, thank you. 

b) Please state what role the funders took in the study. If any authors received a salary from any of your funders, please state which authors and which funder. If the funders had no role, please state: "The funders had no role in study design, data collection and analysis, decision to publish, or preparation of the manuscript." 

Please send your amended statements by return email; we will change the online submission form on your behalf. 

The amended statements have been returned by email, thank you.

[We are grateful to the Gump research station in Moorea (French Polynesia) for letting us work in their station. We are particularly thankful to the directors and the staff for their unconditional help and great friendship. We thank Léonore Bonin for fish capturing. We acknowledge Dr. Christèle Borgeaud for English corrections and valuable comments. We thank Radu Alexandru Slobodeanu for statistical help and support. The study was supported by the Swiss National Science Foundation (grant numbers 31003A_153067 and 310030B_173334/1) to R.B.]

Please remove any funding-related text from the manuscript and let us know how you would like to update your Funding Statement. 

Thank you for your comment. We did remove the funding-related text from the manuscript and updated the Acknowledgments section accordingly on L. 314 - 319.

We did add the captions for the Supporting Information at the end of the manuscript and updated them in the text accordingly, thank you.

Preamble to our replies to the specific referee comments below: we would like to thank the referee for his/her highly useful comments. The different comments have led to substantial changes in the introduction as demanded, and we trust that our revised version make the paper clearer.

Reviewers' comments:

Reviewer's Responses to Questions

Comments to the Author

1. Is the manuscript technically sound, and do the data support the conclusions?

Reviewer #1: Yes

2. Has the statistical analysis been performed appropriately and rigorously?

Reviewer #1: I Don't Know

3. Have the authors made all data underlying the findings in their manuscript fully available?

Reviewer #1: No

Sorry we do not understand, under the ‘Statistical analyses’ part of the manuscript, it is stipulated that ‘All data and the R-scripts to reproduce the analyses are available at https://figshare.com/s/4ed4d7c5bcc750500de9.’ On L. 213-214.

4. Is the manuscript presented in an intelligible fashion and written in standard English?

Reviewer #1: No

5. Review Comments to the Author

Reviewer #1: In this study, the authors tested the ability of cleaner fish to succeed in a simultaneous matching-to-sample task. Such task is classically used to test for rule learning based on a ‘same’ abstract relational concept. This cognitive ability has been established in many animal species from mammals to insects. However, while some fish species appeared able to master this task (zebrafish and goldfish), others were not (e.g. archerfish). The current study thus proposes to further explore fish concept learning capacities by testing an additional species. This question is scientifically valid and of interest as I consider that the comparative cognition field certainly gains from more model species to be tested for key cognitive tasks. The results demonstrate great individual variability but with success of a few individuals. These results support the existence of an ability for solving matching-to-sample task in this species in particular as the procedure chosen is not classic and may have had detrimental effect on performance. Indeed, usually, the sample is not longer present when the subject has to make a choice between a stimulus identical to the sample and an alternative stimulus. In addition, only a limited number of samples are used in standard procedures at least during the acquisition phase while in the current study, each trial presented a novel stimulus. These key differences may have induced attentional bias and noise which prevented most subjects to extract the common rule between trials. Proposing alternative procedures is of high interest to better understand the underlying processes but I regret that the authors did not test in parallel the classical procedure for comparison. With the current data, it is indeed impossible to compare the cleaner fish performances with those of other species.

We would like to thank you for your comments. We agree that we cannot compare our study with others on the simultaneous matching-to-sample task as we did change the procedure substantially. However, we believe that this new procedure might be of great interest for further studies. Indeed, it is known that the use of novel stimuli over trials is the only way to determine whether animals achieved abstract concept learning . According to this view, using the same stimuli repeatedly over trials may only yield indicative results. While some studies have supplemented the initial solving with new stimuli afterwards, on an individual level it could still be a case of ‘learning to learn’. Furthermore, it has been shown that the fact to utilize an increased number of various stimuli enhance the gain of relational ability . We propose that our procedure avoids all these potential shortcomings; if subjects perform above chance it must be based on abstract concept learning. We do agree that to study cleaner fish in the ‘standard’ procedure of simultaneous MTS might be a nice future study. We say so explicitly in the new version in the discussion.

Other comments:

Introduction: The introduction needs quite a substantial editing. Some sentences are difficult to read, some others are incorrect or imprecise. I recommend the authors to invest more effort in drafting the introduction in a novel version of the manuscript.

Thank you for your input. We revised the introduction and hopefully made our writing clearer. Indeed, we changed the order of points and wrote more explicitly about various aspects.

A few examples:

l. 35 : do you mean a shaping procedure ? It is not clear what you mean by operant conditioning. Your description does not allow to understand what is the procedure to study the ability of animals to learn abstract concepts. Please revise the lines 34-37

Thank you for your comment. We are now much more explicit about the experimental paradigm, which should make things clear on L. 57 – 63.

l. 46 : I do not understand this sentence. ‘concepts are represented by the relation of abstract rules to novel stimuli’ ??

Thank you for your comment. We chance the sentence accordingly on L. 50 - 53.

l. 49-53 : Difficult to follow, please revise for improved clarity

We change this sentence accordingly to clarify the procedure on L. 78 – 80. Thank you

l.57 : The reference 27 seems fully out of context here. Could you please provide references of studies using simultaneous MTS to test for same/difference concept learning ?

We apologize, the right references were added on L. 75. Thank you

l. 58: Incorrect references as they used a DMTS procedure and not a simultaneous MTS

…

Thank you for your remark. We change the reference accordingly on L. 75.

Methods

l. 164 : Why did you feed the fish ab libitum before the experiment. This should induce a lack of motivation for the food reward.

Thank you for your comment. ‘We changed the word ‘ad libitum’ as we simply offered food for 5 min. Cleaner fish eat 11 hours during the day and hence have an adapted digestion. This implies that they are hungry in the morning, after 13h without food. The goal of giving them some food prior to the start of the trials was to avoid that they are so hungry that they do not pay attention to the stimuli. The 10 trials per day were spread over 5-6 h, so any lack of motivation due to the feeding would have affected at best the first trial (L. 191 – 197).

l. 165 : Why did you let the food plate in place for 2 hours ?

I suggest to provide a video as supporting information to help the reader to better picture the whole procedure.

Thank you for your comment. We left the plate for about 2 hours after the trials to make sure that fish did eat enough for the day (L. 191 - 197). Food intake during trials was quite low.

Discussion

l.228 : There is at least one study, the one in pigeons which is described a few sentences later (ref 12)

Thank you for your remark. In the study with the pigeons, they did use 76 stimuli but those stimuli were used every day. Here in our study, no same stimuli were shown over the 200 trials (so 400 stimuli were used in total without being used twice) (L. 269 – 271).

Bibliography : please check the references formatting as not consistent between references and does not follow the journal guidelines.

Thank you for your comment. The bibliography was carefully checked and the formatting respected to follow the journal guidelines (Vancouver style) on L. 323 and so on.

6. PLOS authors have the option to publish the peer review history of their article (what does this mean?). If published, this will include your full peer review and any attached files.

Do you want your identity to be public for this peer review? For information about this choice, including consent withdrawal, please see our Privacy Policy.

Reviewer #1: No

---

## [Decision Letter · Decision Letter 1]

22 Dec 2021

Cleaner wrasse Labroides dimidiatus perform above chance in a "Matching-to-sample" experiment

PONE-D-21-17431R1

Dear Dr. Aellen,

We’re pleased to inform you that your manuscript has been judged scientifically suitable for publication and will be formally accepted for publication once it meets all outstanding technical requirements.

Kind regards,

Adrian G Dyer, Ph.D.

Academic Editor

PLOS ONE

Additional Editor Comments (optional):

Appologies for slow response; 2021 is a difficult year for obtaining reviewers as many people are disrupted. The manuscript has now been reviewed twice by a world expert on the topic, and I have also carefully read the manuscript versions. As the expert reviewer now recommends acceptance; I am recommending acceptance of the manuscript.

Reviewers' comments:

Reviewer's Responses to Questions

**Comments to the Author**

1. If the authors have adequately addressed your comments raised in a previous round of review and you feel that this manuscript is now acceptable for publication, you may indicate that here to bypass the “Comments to the Author” section, enter your conflict of interest statement in the “Confidential to Editor” section, and submit your "Accept" recommendation.

Reviewer #1: All comments have been addressed

2. Is the manuscript technically sound, and do the data support the conclusions?

Reviewer #1: Yes

3. Has the statistical analysis been performed appropriately and rigorously? 

Reviewer #1: Yes

4. Have the authors made all data underlying the findings in their manuscript fully available?

Reviewer #1: Yes

5. Is the manuscript presented in an intelligible fashion and written in standard English?

Reviewer #1: Yes

6. Review Comments to the Author

Reviewer #1: (No Response)

7. PLOS authors have the option to publish the peer review history of their article (what does this mean?). If published, this will include your full peer review and any attached files.

Reviewer #1: No

---

## [Editor Report · Acceptance letter]

21 Jan 2022

PONE-D-21-17431R1 

Cleaner wrasse *Labroides dimidiatus* perform above chance in a “Matching-to-sample” experiment 

Dear Dr. Aellen:

I'm pleased to inform you that your manuscript has been deemed suitable for publication in PLOS ONE. Congratulations! Your manuscript is now with our production department. 

Kind regards, 

on behalf of

Dr. Adrian G Dyer 

Academic Editor

PLOS ONE